# Group arts therapies for patients with schizophrenia: a protocol of systematic review and meta-analysis

Aijia Zhang [ID],[1] Xuexing Luo [ID],[1] Runqing Lin,[1] Caihong He,[2] Jue Wang [ID],[3,4,5] Guanghui Huang[1,6]

AZ and XL contributed equally.

AZ and XL are joint first authors.

For numbered affiliations see end of article.

**Correspondence to**
Associate Professor Guanghui Huang;
ghhuang1@must.edu.mo and Assistant Professor Jue Wang; wangjue@must.edu.mo

## ABSTRACT

**Introduction** Schizophrenia, a chronic mental problem, significantly impacts cognition, emotion and social functioning. Conventional pharmacotherapy faces challenges including numerous side effects, low adherence to medication and substantial costs. In this context, group arts therapies (GATs) emerge as a promising complementary approach for symptom alleviation in schizophrenia patients. Nonetheless, the effectiveness and safety of GATs are yet to be firmly established. This study aims to systematically assess the therapeutic impact of all group-based artistic interventions as complementary treatments for schizophrenia, focusing on their potential benefits.

**Methods and analysis** This study will search four English-language databases (PubMed, Web of Science, Cochrane Library and Embase), two Chinese databases (Wanfang Data and China National Knowledge Infrastructure) and three Korean databases (RISS, Korean Citation Index and DBpia) from their inception until October 2023. It will include all randomised controlled trials that compare GATs for schizophrenia with standard rehabilitation methods. The primary outcome is the improvement in patients' positive and negative symptoms. Methodologies such as bias risk assessment, data synthesis, sensitivity analysis and subgroup analysis will be implemented using Review Manager V.5.4. Study results with high heterogeneity will be merged using a random-effects model ($I^2$>50% or p<0.1). In cases where meta-analysis is not viable due to significant clinical and methodological heterogeneity, a qualitative summary of the findings will be provided.

**Ethics and dissemination** The data used in this systematic review are anonymised, devoid of any private information, eliminating the requirement for ethical approval. Dissemination of the research findings will be conducted via peer-reviewed publications.

**PROSPERO registration number** CRD42023471583.

## STRENGTHS AND LIMITATIONS OF THIS STUDY

⇒ This systematic review will incorporate diverse modalities such as painting, music, dance, yoga, drama, handicrafts, collage, play, photography, and movie.
⇒ The study displays relatively minor language bias, as research published in Chinese and Korean has been included alongside English-language studies.
⇒ Potential publication bias may exist due to the exclusion of conference papers, unpublished documents and ongoing clinical trials from the included literature.
⇒ Significant heterogeneity might be present owing to variations in the type of group art therapy, as well as differences in the duration and frequency of the treatment.

## INTRODUCTION

Individuals with schizophrenia endure significant physiological and psychological stress. A recent epidemiological survey shows that these individuals face a higher risk of suicide and unemployment compared with the general population.[1 2] This increased risk is attributed to the presence of positive symptoms such as hallucinations, delusions and disordered thinking.[3] Furthermore, physiological distress intensifies the feelings of anxiety and depression, potentially leading to dire decisions.[4] The condition is also characterised by negative symptoms, including cognitive impairments, emotional flatness, social withdrawal and diminished motivation, which severely limit the ability to participate in social life. These symptoms are the primary causes of their long-term incidence rate and may struggle to undertake the activities of daily living.[5–7]

At present, pharmacotherapy serves as the principal treatment approach for schizophrenia, showing relative efficacy in mitigating positive symptoms and preventing relapse.[8 9] However, studies suggest that medication alone has limited effectiveness in reducing negative symptoms.[10] Furthermore, the dire side effects of long-term medication use, poor patient compliance and the prohibitive costs of treatment present significant challenges for many patients and their families.[11–15] Thus, it is imperative to offer an adjunctive therapeutic option from the perspective of psychotherapy to assist

individuals with schizophrenia to live a fulfilling life alongside their condition.

Group arts therapies (GATs) represent a novel psychotherapeutic approach using the collective creation of art under professional guidance to support mental health.[16–18] Employing mediums like painting, music, dance and games, GATs focus on the composition and interaction of group members to boost mental well-being.[19–21] Certain studies affirm the significant impact of GATs on the social functioning deficits,[22] sense of pleasure,[23] self-efficacy[24] and interpersonal relationships[25] of individuals with schizophrenia. The therapeutic benefits stem partly from the diverse artistic mediums that redirect patients' focus to the process of creation, helping to alleviate anxiety and stress, thus facilitating emotional expression and ameliorating the presentation of positive symptoms.[26] Furthermore, by cultivating artistic skills, GATs improve self-confidence and emotional, cognitive and social functioning.[27] The interactive nature of GATs also promotes physiological, psychological and social well-being through shared experiences and support within the therapy group.[28] For individuals facing verbal communication challenges, GATs provide a platform for unencumbered emotional expression, potentially more effective than traditional verbal therapies.[29] Listening to others share their artworks can help relieve negative symptoms.[30 31] The advantage is especially evident in community settings, where collaborative art-making fosters meaningful interactions. The interaction can help these individuals overcome resistance and enhance their positive identity and personal recovery.[32 33] Arts therapies in the community can also help in reducing the stigma of mental illness.[34 35]

Despite these benefits, a systematic review of GATs' effectiveness in treating schizophrenia is lacking. This review aims to fill the gap by evaluating the evidence for GATs in clinical settings and offering a comparative analysis not present in the existing literature.

## METHODS
### Criteria for including studies in the review
#### Types of studies
All randomised controlled trials focused on patients diagnosed with schizophrenia will be included in the study. Scholarly journals published in Chinese, English and Korean will constitute the primary publication types considered for inclusion. Duplicated publications, unpublished data, studies with inaccessible complete information, non-experimental reports, observational studies, conference papers, literature reviews, case study reports, lectures or critiques involving outcomes unrelated to GATs' interventions will be excluded from this research.

#### Types of participants
Participants aged 18–70 years old, regardless of gender, ethnicity, nationality, educational level or economic status,

will be incorporated into the study. Diagnoses will be based on the Diagnostic and Statistical Manual of Mental Disorders (DSM)-III,[36] DSM-III-R,[37] DSM-IV[38] International Classification of Diseases–10 criteria[39] or any other recognised standards. Individuals with significant organic injury, functional impairments, coexisting psychiatric conditions or those who are unable to continue observation due to severe physical illness, as well as participants with incomplete clinical data, will be excluded from the study.

#### Types of interventions
Experiments comparing group-based arts therapies' interventions with no treatment, standard pharmacotherapy, routine care or any other form of active psychological counselling intervention will be included. Additionally, studies examining combined interventions, such as GATs in conjunction with standard pharmacotherapy, compared with solely using an alternative modality, such as standard pharmacotherapy alone, also fall within the scope of this research.

Specifically, the British Association of Art Therapists (BAAT) definition of arts therapies was used as inclusion criteria for the intervention. BAAT proposes that arts therapies require participants to use art materials for self-expression and reflection in the presence of a trained art therapist.[40] Therefore, the types of GATs that will be included in the study include painting, music, dance, yoga, drama, games, craft making, collage, photography and film. Another necessary requirement is the involvement of qualified professionals such as art therapists or doctors during the research process. Additionally, the settings in which interventions are delivered include inpatient, outpatient and community settings. Furthermore, because in addition to necessary treatment settings such as hospitals and institutions, the community provides key rehabilitation intervention settings for people with schizophrenia that help improve social skills and quality of life, the settings in which interventions are implemented will include inpatient, outpatient and community.[41 42] Given the focus on arts therapies methods conducted within a group or community setting, individualised intervention measures will be excluded.

#### Types of outcomes
The primary outcomes will be the improvement of positive and negative symptoms in individuals with schizophrenia, to be measured using validated instruments such as the Positive and Negative Syndrome Scale.

Secondary outcomes will encompass functional improvements associated with GATs, including assessments of daily living capabilities, social functioning, emotional functioning and cognitive functioning, as well as adverse events, compliance, costs and cost-effectiveness. These outcomes will be evaluated using publicly recognised scales such as the Generic Quality of Life Inventory-74, Personal and Social Performance

**Table 1** Search strategies.

| Number | Search terms |
| --- | --- |
| #1 | Group Arts therapies |
| #2 | Group Painting Therapy |
| #3 | Group Music Therapy |
| #4 | Group Dance Therapy |
| #5 | Group Yoga Therapy |
| #6 | Group Drama Therapy |
| #7 | Group Handicrafts Therapy |
| #8 | Group Collage Therapy |
| #9 | Group Play Therapy |
| #10 | Group Photography Therapy |
| #11 | Group Movie Therapy |
| #12 | #1 OR #2 OR #3 OR #4 OR #5 OR #6 OR #7 OR #8 OR #9 OR #10 OR #11 |
| #13 | Schizophrenia, schizophrenias |
| #14 | Schizophrenic Disorders |
| #15 | Schizophrenic Disorder |
| #16 | Disorder, Schizophrenic |
| #17 | Disorders, Schizophrenic |
| #18 | #13 OR #14 OR #15 OR #16 OR #17 |
| #19 | #12 AND #18 |

Scale, Nurses' Observation Scale for Inpatient Evaluation and Tennessee Self-Concept Scale, among others.

### Electronic searches

Two researchers will conduct a search for all relevant literature published up to October 2023 in English, Chinese and Korean. The literature will be sourced from databases such as PubMed, Web of Science, Cochrane Library, Embase, Wanfang Data, China National Knowledge Infrastructure, RISS, Korean Citation Index and DBpia. The keyword combinations used will include (a) group arts therapies and (b) schizophrenia. The search strategy for PubMed is illustrated in table 1 and online supplemental appendix 1. Equivalent search strategies will be used in Chinese databases and Korean databases.

### Searching other resources

The reference lists of published relevant academic journals and review articles will also be manually searched to identify any missing literature. Furthermore, clinical guidelines pertaining to the relevant trials and articles of an ambiguous nature will be manually scrutinised to ensure no eligible studies are overlooked.

### Projected timeline for the review

The systematic review is slated to commence in January 2024 and is envisioned to reach completion by June 2024.

## DATA COLLECTION

### Selection of studies

Two independent researchers will independently review the titles, abstracts and full texts of the articles retrieved, compiling studies that meet the inclusion criteria. Any disagreements between researchers will be resolved through discussion to reach a consensus, and if necessary, a third independent reviewer will contribute to the final decision. The flowchart of the Preferred Reporting Items for Systematic Reviews and Meta-Analyses delineates the selection process of the studies (figure 1).

### Data extraction and management

Two researchers will independently extract the following information from the included articles using a standardised form[1]: basic information—first author's name, year of publication, region of publication, number of cases included and type of study design[2]; clinical and pathological data: age, gender, duration of illness, length of hospital stay and inpatient environment[3]; intervention measures: nature of the intervention, implementer, frequency of intervention per week and duration of the intervention[4]; outcome measures[5] and rating scales used. Both researchers will independently extract and code data according to this form. The results of the data extraction will be cross-checked. In case of discrepancies, a third researcher will be involved in the discussion and decision-making process.

### Assessment of risk of bias

The risk of bias and methodological quality of all included randomised controlled trials will be assessed by two independent researchers using the Cochrane Risk of Bias 2 tool.[43] Each trial will be evaluated for bias according to the following domains: bias arising from the randomisation process, bias due to deviations from the intended interventions, bias due to missing outcome data, bias in the measurement of the outcome and bias in selection of the reported result. Risks of bias will be classified as low, high or unclear. In cases of disagreement, a third reviewer will be consulted, and consensus will be reached through discussion.

## DATA ANALYSIS

### Measures of treatment effect

Review Manager (RevMan) V.5.4 will be used to perform the meta-analysis. Continuous data will be expressed as mean difference and dichotomous data as risk ratio, both with 95% CI. When the same outcome is measured in various ways, the standardised mean difference with a 95% CI will be chosen to represent continuous data.

### Unit of analysis issues

The primary unit of analysis will be all individuals included in the control and experimental groups. Units of outcome

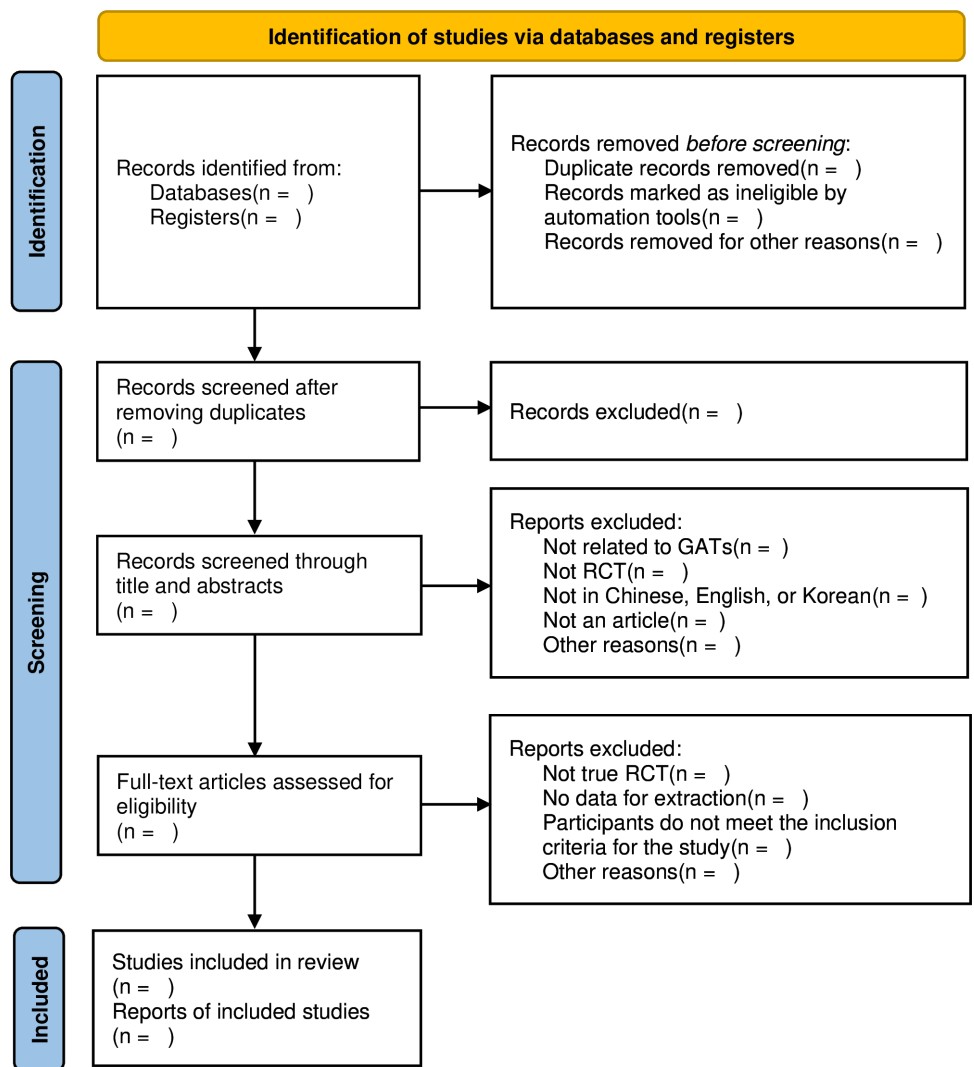

**Figure 1** PRISMA flow diagram of the study process. GATs, group arts therapies; PRISMA, Preferred Reporting Items for Systematic review and Meta-Analysis; RCT, randomized controlled trial.

measures from different studies will be converted to the International System of Units prior to analysis.

### Dealing with missing data

Researchers will attempt to contact the authors included in the study via email to obtain missing or incomplete data. If missing data cannot be obtained, the study will be excluded.

### Assessment of heterogeneity

Statistical heterogeneity among the included studies will be assessed using the $I^2$ statistics and the $\chi^2$ test. Studies will be considered heterogeneous when $I^2 > 50\%$ or $p < 0.1$.

### Assessment of reporting biases

Reporting bias will be assessed following the CONSORT guidelines, and a funnel plot will be generated to evaluate potential reporting biases. A symmetric distribution of data within the funnel plot suggests the absence of publication bias. If the funnel plot is asymmetrical, efforts will be made to explore possible explanations beyond publication biases and language biases.

### Data synthesis

Data synthesis will be conducted using RevMan V.5.4. Results from two or more heterogeneous studies will be combined using a random effects model ($I^2 > 50\%$ or $p < 0.1$). In the event that the meta-analysis is impeded by considerable clinical and methodological heterogeneity, a qualitative summary of the results will be provided.

### Subgroup analysis and investigation of heterogeneity

In the presence of significant heterogeneity among the included studies, subgroup analyses will be conducted. Subgroups will be formed based on different populations (eg, male vs female), different types of GATs mediums (such as group painting, group music or group play) and different art therapy settings (such as inpatient, outpatient or community). The effect estimates for each subgroup will be calculated and compared inter-subgroup.

### Sensitivity analysis

To ensure the robustness of the evidence, sensitivity analyses will be performed to assess the impact of

studies with a high risk of bias and to explore possible sources of heterogeneity. Decisions to exclude lower-quality studies will be made by comparing the outcomes and considering factors such as sample size, strength of evidence and their influence on the pooled effective sample size.

## Evidence quality evaluation

The quality of evidence will be further assessed using the Grading of Recommendations Assessment, Development and Evaluation approach, which categorises the quality of evidence into four levels: very low, low, moderate or high. In interpreting the findings, consideration will also be given to the quality of evidence, the context of the studies, potential benefits and harms and patient values.

## Patient and public involvement

Patients or the public were not involved in the study.

## Ethics and dissemination

This systematic review does not require ethical approval, as all data used are anonymous and do not involve any private information. The outcomes of the research will offer evidence on the efficacy and safety of GATs as an adjunctive therapeutic approach for schizophrenia and will be disseminated through peer-reviewed publications.

## MODIFICATIONS TO THE PRELIMINARY PROPOSAL

Three significant modifications were made to the original protocol in response to reviewers' feedback. First, a clear definition for GATs, based on the BAAT definitions, was provided to ensure alignment of the reviewed interventions with standard professional guidelines. Second, the criteria for the studies to be included in this review were detailed further. The revised standards now encompass a broader array of art therapies modalities such as painting, music, dance, yoga, drama, handicrafts, collage, play, photography and movie. Moreover, the inclusion criteria were expanded to accommodate studies carried out exclusively in community settings. Finally, recognising the inherent variability in study designs and participant demographics within GATs, the expectation of low between-study heterogeneity may be overly optimistic. As such, the research methods clearly articulate how the statistical heterogeneity of the included studies was evaluated using the $I^2$ statistics and the $\chi^2$ test.

## DISCUSSION

While some explorations of GATs in the field of schizophrenia have been undertaken, evidence of their effectiveness in ameliorating positive, negative and other related symptoms in patients remains limited. The inaugural MATISSE study on GATs indicated that GATs do not improve the overall functioning or mental health of individuals with schizophrenia,[44] casting doubt on the overall efficacy of GATs. Since then, researchers have been seeking evidence to support the mechanistic changes induced by GATs. Evidence from randomised controlled trials of group painting therapy,[23] group music therapy,[45] group play therapy[46] and group dance therapy[47] indicates that group forms of treatment using different art media can effectively improve a variety of clinical outcomes in patients with schizophrenia. Additionally, a large-scale study is examining GATs' effectiveness in a diagnostically diverse patient population within mental health services,[20] yielding promising initial findings. However, these investigations have not systematically examined the efficacy of different GAT modalities.

A review study on arts therapies for schizophrenia was conducted in 2005.[48] But due to the early start of the study, only two randomised controlled trials with insufficient participants were included, making the results less meaningful. A meta-analysis initiated in July 2021 intended to assess arts therapy effectiveness for schizophrenia failed to report outcomes. Uttley *et al*'s systematic review highlighted the potential cost-effectiveness of GATs for non-psychotic disorders, yet noted the evidence was too limited for a definitive comparison.[49] Reviews on drama therapy,[50] dance therapy[51] and music therapy[52] examined the benefits and harms of different types of arts therapies for patients with schizophrenia. However, similar to previous studies, there is only moderate to low-quality evidence that arts therapies can be used as a supplement to standard care to improve relevant clinical outcomes in schizophrenia. At the same time, these efforts are not conducted in groups or small groups. Whether subgroups such as type, medium, duration or frequency of GATs are related to treatment effectiveness remains unknown. The inclusion of large-scale and high-quality evidence and subgroup analyses based on these variables may help address this issue.

This study aims to systematically assess the effectiveness of group-based artistic interventions in clinical treatment for schizophrenia, offering insights not provided by prior reviews. By examining a broader range of studies, including those published in English, Chinese and Korean, this review seeks to enhance clinical evidence for healthcare providers and patients considering GATs as supplementary schizophrenia treatment.

Potential limitations of this study include possible language bias from excluding research in languages other than English, Chinese and Korean, and publication bias due to the omission of conference papers, unpublished literature and ongoing trials. Additionally, variability in GATs' types, mediums, durations and frequencies may introduce heterogeneity. Nonetheless, this systematic approach still contributes to a broader understanding of GATs for schizophrenia.

**Author affiliations**
[1]Faculty of Humanities and Arts, Macau University of Science and Technology, Taipa, Macau, China
[2]Operation Centre, Guangzhou Wanqu Cooperative Institute of Design, Guangzhou, Guangdong, China
[3]State Key Laboratory of Quality Research in Chinese Medicine, Macau University of Science and Technology, Taipa, Macau, China
[4]Faculty of Chinese Medicine, Macau University of Science and Technology, Taipa, Macau, China
[5]Guangdong-Hong Kong-Macao Joint Laboratory for Contaminants Exposure and Health, Guangdong University of Technology, Guangzhou, Guangdong, China
[6]Zhuhai MUST Science and Technology Research Institute, Zhuhai, Guangdong, China

**Contributors** AZ wrote the draft of the manuscript; XL and JW did independent reviews; RL researched the references; CH, AZ and RL contributed to data organisation and JW and GH critically reviewed the final draft of the manuscript. All authors contributed to the article and approved the submitted version.

**Funding** This study is supported by the Research on Digital Art and Cultural Industry Development project of Guangzhou Wanqu Cooperative Institute of Design (9028) and Macau University of Science and Technology's Faculty Research Grant (No.: FRG-24-049-FA).

**Competing interests** None declared.

**Patient and public involvement** Patients and/or the public were not involved in the design, conduct, reporting or dissemination plans of this research.

**Patient consent for publication** Not applicable.

**Provenance and peer review** Not commissioned; externally peer reviewed.

**ORCID iDs**
Aijia Zhang http://orcid.org/0009-0009-1435-6319
Xuexing Luo http://orcid.org/0000-0001-9384-5931
Jue Wang http://orcid.org/0000-0002-6151-1117

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
