## [Reviewer comments · BMJ Open]

ARTICLE DETAILS

TITLE (PROVISIONAL)	Group Arts Therapies for Patients with Schizophrenia: A Protocol of Systematic Review and Meta-analysis
AUTHORS	Zhang, Aijia; Luo, Xuexing; Lin, Runqing; He, Caihong; Wang, Jue; Huang, Guanghui

VERSION 1 – REVIEW

REVIEWER	Khodaei-Ardakani, Mohammad-Reza University of Social Welfare and Rehabilitation Sciences, Psychiatry
REVIEW RETURNED	07-Dec-2023

GENERAL COMMENTS	Thanks for the systematic review. comments:  -A precise and scientific definition of art therapy has not been provided. - In some reviewed articles, a complete comparison between conventional treatments and art therapy has not been done and is not double-blind. -The number of reviewed articles is limited and cannot be generalized to the whole. -Entry criteria should be wider and more diverse.
---

REVIEWER	Millard, Emma Queen Mary University of London, Unit for Social and Community Psychiatry
REVIEW RETURNED	02-Jan-2024

GENERAL COMMENTS	Thoughts  • Very useful and much needed • Well-written and clear • Language could be more inclusive – ‘disorder’ doesn’t really fit with recovery-focused models of treatment. I would recommend that you read Leamy (2011), as this seems more in line with the arts therapies and their approach to supporting people with severe mental illness, than the medical model and language that you are using. o Intro – paragraph 1 - Saying that people with schizophrenia cannot live as ‘ordinary people’ is also not inclusive language, could you say ‘may struggle to undertake activities of daily living’ instead? o Intro – paragraph 2 – ‘integration into normal life’ is also problematic. Could you change to ‘reaching their full potential’ or something like that? The recovery model suggests that the aims of treatment for chronic mental health conditions should not be to make people ‘normal’, it should be to support them to live a fulfilling life alongside their condition. • Another well-documented benefit of group arts therapies, that you
--

	could include in your introduction, is that it can be a stepping stone into increased community engagement. This is likely because of increased confidence and a reminder of how helpful community participation can be for wellbeing.  • You have not included dramatherapy in your list of arts therapies – this is very much included under the ‘arts therapies’ umbrella in the UK. • It would be useful for you to state whether you will include both inpatient and community settings in your included studies. • You have not stated how you will define what counts as the arts therapies, will you require the studies to be delivered by a qualified therapist? Or will they be self-defined as being ‘arts therapies’? This needs to be really clear as there are such big differences between countries as to how they define arts therapies. • I am not familiar with Unit of Analysis considerations, but I wonder if yours will be the groups of people, rather than individuals, as you are analysing group data rather than individual-level data. • I am confused by your decision about the type of model you will be using – are you planning to split the studies between those that have ‘high’ heterogeneity vs ‘low’ heterogeneity? You will be looking for heterogeneity between studies, so how will you know which clusters of studies have high/low heterogeneity? I would suggest that you are unlikely to find studies with low heterogeneity, so a random-effects model for all of them would be more appropriate. • I would like to know more about why you have not involved patients or the public. It is best practice to seek input from the population you are writing about. • In your discussion you write about MATISSE. You say it was a ‘systematic’ study, which is a bit confusing, as it was a trial not a review. You might want to adjust your language to make this clear. It was also only a trial of art therapy, not multiple arts therapies. • There is currently a large-scale trial of group arts therapies being undertaken in the UK – the ERA study. There is a protocol paper which you might be interested in – Carr (2023) Effectiveness of group arts therapies (art therapy, dance movement therapy and music therapy) compared to group counselling for diagnostically heterogeneous psychiatric community patients: study protocol for a randomised controlled trial in mental health services (the ERA study) • There is a Cochrane review about art therapy for schizophrenia from 2005, one about music therapy from 2017, one about dance therapy from 2013, and one about dramatherapy from 2007. These should certainly be discussed.
--	--

VERSION 1 – AUTHOR RESPONSE

Response to Reviewer #1

We appreciate reviewers’ time and effort in reviewing our manuscript (bmjopen-2023-082076- “Group Arts Therapies for Patients with Schizophrenia: A Protocol of Systematic Review and Meta-analysis”), and your valuable comments and professional advice have greatly helped us improve the quality of our manuscript. We have carefully addressed all the issues raised and have modified the paper accordingly in “Main Document - marked copy” (highlighted in YELLOW and labeled by Microsoft Word’s Track Changes tool).

Please find below a detailed point-by-point response to all comments (the reviewer’s comments in BLACK, our replies in BLUE). It would be greatly appreciated if you could review and reconsider our revised manuscript!

Comment 1. A precise and scientific definition of art therapy has not been provided.

Response:

Thanks for your advice!

In response to your valuable feedback, we have made concerted efforts to address this concern by elaborating on our understanding and interpretation of group arts therapies. Acknowledging the absence of a universally accepted academic definition for group arts therapies, we have synthesized insights from prior research to construct a comprehensive description that accurately reflects its application and objectives within the scope of our study. This elaboration can be found in lines 142-145 of the revised manuscript.

Furthermore, recognizing the importance of grounding our discussion in established professional standards, we have incorporated the definition of arts therapies as provided by the British Association of Art Therapists (BAAT) into lines 263-268. We believe that quoting this definition aids in clarifying the type of intervention under review and aligns our discourse with recognized professional guidelines.

Comment 2. In some reviewed articles, a complete comparison between conventional treatments and art therapy has not been done and is not double-blind.

Response:

We deeply appreciate your insightful suggestion highlighting the necessity of addressing the limitations inherent in some of the reviewed articles, particularly the lack of complete comparisons between traditional therapy and arts therapies, and the absence of double-blind protocols. Your observation has been instrumental in guiding our efforts to refine our manuscript for a more rigorous and nuanced analysis.

In response to your valuable feedback, we have undertaken several modifications to our manuscript, aimed at addressing these concerns:

Firstly, we have explicitly defined the scope of the interventions assessed in our review (lines 254–259) to ensure that all included studies offer a direct comparison between the efficacy of traditional treatments and group art therapy. This adjustment allows us to more accurately gauge the therapeutic value of art therapy in relation to conventional treatment modalities.

Furthermore, acknowledging the limitations arising from non-double-blinded studies, we have committed to a systematic review of all extracted information using standardized forms. This process encompasses a range of crucial data points, including basic study information, clinical pathological data, interventions, outcome indicators, and rating scales (lines 352-361). This structured approach enhances our assessment's thoroughness and reliability.

To improve the methodological rigor of our review further, we have also integrated the use of the Cochrane Risk of Bias 2 (RoB 2) tool into our evaluation process. By applying this tool (lines 363-370), we aim to ensure the inclusion of higher-quality literature while identifying and addressing potential biases and methodological issues within these studies.

Comment 3. The number of reviewed articles is limited and cannot be generalized to the whole.

Response:

Thank you for your insightful comment regarding the number of articles included in our review. We acknowledge the concern that the limited number of reviewed articles may impact the generalizability of our findings.

We have implemented several pivotal changes to address the possible limitations associated with the initial study sample size:

We have broadened our search parameters, which now incorporate a wider spectrum of search terms. This strategic enhancement is detailed in lines 294-297 of the revised manuscript, ensuring that additional relevant studies are considered for inclusion in our review.

In anticipation of potential heterogeneity among the included studies, we have provisioned for subgroup analyses as part of our methodology (lines 414-419). This will enable us to delve into whether certain modalities or methodologies within group art therapy demonstrate differing levels of effectiveness. Such analyses can offer valuable specificity regarding which interventions may yield the most benefit for patients with schizophrenia.

We will also introduce a comprehensive discussion section that rigorously examines both the strength and reach of our findings (lines 411-412).

Comment 4. Entry criteria should be wider and more diverse.

Response:

We greatly appreciate your valuable suggestion to widen and diversify our entry criteria for the review. To align with your recommendation, we have expanded our inclusion criteria by integrating the definition of art therapy provided by the British Association of Art Therapists (BAAT). This definition enables us to encapsulate a wider variety of therapeutic modalities within our review, beyond what was initially considered. Consequently, our revised criteria now encompass an array of art therapy genres, including painting, music, dance, yoga, drama, games, craft making, collage, photography, and film. This addition is articulated in lines 263-268 of the revised manuscript. Furthermore, in response to your feedback regarding the need for a clear and comprehensive search strategy, we have meticulously detailed our search methodology in Table 1 (line 297) of our manuscript. We are grateful for your constructive critique, which has undeniably contributed to enhancing the quality and integrity of our work. We hope that our revisions have adequately addressed your concerns.

Response to Reviewer #2

We appreciate reviewers' time and effort in reviewing our manuscript (bmjopen-2023-082076- "Group Arts Therapies for Patients with Schizophrenia: A Protocol of Systematic Review and Meta-analysis"), and your valuable comments and professional advice have greatly helped us improve the quality of our manuscript. We have carefully addressed all the issues raised and have modified the paper accordingly in "Main Document - marked copy" (highlighted in BLUE and labeled by Microsoft Word's Track Changes tool). Please find below a detailed point-by-point response to all comments (the reviewer's comments in BLACK, our replies in BLUE). It would be greatly appreciated if you could review and reconsider our revised manuscript!

Comment 1. Language could be more inclusive – 'disorder' doesn't really fit with recovery-focused models of treatment. I would recommend that you read Leamy (2011), as this seems more in line with the arts therapies and their approach to supporting people with severe mental illness, than the medical model and language that you are using.

Response:

Thank you for your insightful feedback on the language and perspective presented in our manuscript. Following your recommendation, we have reviewed Leamy et al (2011) and have found it to be highly informative, particularly in advocating for a paradigm that emphasizes personal recovery. Inspired by this, we have revised our manuscript to reflect a more person-centered approach that aligns with the ethos of arts therapies. We have made efforts to replace the term 'disorder' with more suitable language where appropriate throughout the text (e.g. line 4).

Comment 2. Intro – paragraph 1 - Saying that people with schizophrenia cannot live as 'ordinary people' is also not inclusive language, could you say 'may struggle to undertake activities of daily living' instead?

Response:

Thank you for your valuable recommendation to employ more inclusive language in our introduction. The original phrasing that suggested people with schizophrenia cannot live as 'ordinary people' was indeed not appropriately inclusive. In response to your suggestion, we have carefully revised the language in lines 117-118 of our manuscript. We have amended this sentence to read: "These symptoms are the primary causes for their long-term incidence rate and may struggle to undertake activities of daily living".

Comment 3. Intro – paragraph 2 – 'integration into normal life' is also problematic. Could you change to 'reaching their full potential' or something like that? The recovery model suggests that the aims of treatment for chronic mental health conditions should not be to make people 'normal', it should be to support them to live a fulfilling life alongside their condition.

Response:

We appreciate your thoughtful input regarding the language used in the second paragraph of our introduction. In accordance with your suggestion, we have revised the text in lines 139-141. Our updated sentence now reads: "It is imperative to offer an adjunctive therapeutic option from the perspective of psychotherapy to assist individuals with schizophrenia to live a fulfilling life alongside their condition".

Comment 4. Another well-documented benefit of group arts therapies, that you could include in your introduction, is that it can be a steppingstone into increased community engagement. This is likely because of increased confidence and a reminder of how helpful community participation can be for wellbeing.

Response:

Thanks for your advice!

We have enriched the introduction section to include an in-depth discussion on the benefits of group arts therapy in promoting community engagement (lines 157-162). This addition aims to underline the broader impact of such therapeutic interventions beyond individual symptom management, highlighting their potential for fostering social connections and community integration.

Comment 5. You have not included dramatherapy in your list of arts therapies – this is very much included under the 'arts therapies' umbrella in the UK.

Response:

Thank you for emphasizing the significance of including a comprehensive range of interventions under the umbrella of 'arts therapies', and for pointing out the omission of drama therapy in our initial manuscript.

To address this, we enriched the inclusion criteria based on the British Association of Art Therapists (BAAT) definition of arts therapies. This expansion now notably incorporates drama therapy alongside other therapeutic interventions such as yoga, craft making, collage, photography, and film. These modifications are detailed in line 267 of our manuscript. Correspondingly, we updated our PubMed search strategy in Table 1 (line 297) to reflect these modifications, ensuring a comprehensive capture of relevant literature across these varied modes of creative therapeutic engagement.

Comment 6. It would be useful for you to state whether you will include both inpatient and community settings in your included studies.

Response:

Thank you for your invaluable suggestion regarding the clarity of the settings in which the studies for our systematic review and meta-analysis will be included. Given that in the Introduction we added a discussion of the potential advantages of implementing group arts therapies for people with schizophrenia in community settings (lines 157-162), in the methods and data analysis sections we explicitly added inpatient and community settings in the criteria for including studies.

We have expanded our inclusion criteria to specifically encompass studies conducted in community settings (lines 271-275). This amendment ensures that our systematic review and meta-analysis comprehensively capture the varied contexts in which group arts therapies are implemented.

In addition, we will perform a subgroup analysis of community settings in the presence of high heterogeneity across studies (lines 417-418). This approach will allow us to specifically investigate the nuances of these interventions within community-based contexts.

Comment 7. You have not stated how you will define what counts as the arts therapies, will you require the studies to be delivered by a qualified therapist? Or will they be self-defined as being 'arts therapies'? This needs to be really clear as there are such big differences between countries as to how they define arts therapies.

Response:

We sincerely appreciate your constructive feedback regarding the necessity for a clear and precise definition of group arts therapies in our study. Acknowledging the absence of a universally accepted academic definition for group arts therapies, we have synthesized insights from prior research to construct a comprehensive description that accurately reflects its application within the scope of our study (lines 142-145).

Additionally, we have adopted the definition provided by the British Association of Art Therapists (BAAT) to refine our inclusion criteria for studies. As stated in lines 271–275, this definition emphasizes that true art therapy is facilitated by qualified professionals who guide participants in using art materials for self-expression and reflection. Consequently, we are only including studies in our review that involve interventions conducted by qualified art therapists or similarly credentialed medical professionals.

Comment 8. I am not familiar with Unit of Analysis considerations, but I wonder if yours will be the groups of people, rather than individuals, as you are analysing group data rather than individual-level data.

Response:

We deeply appreciate your insightful query concerning our choice of Unit of Analysis within our study. While our research does indeed encompass data pertaining to groups in the context of group arts therapies, we recognize, as you astutely noted, that a more granular approach is customary in meta-analysis practices (<https://doi.org/10.1136/bmjopen-2017-019066>; <https://doi.org/10.1136/bmjopen-2015-010866>). Such an approach typically focuses on individual-level data, an established practice that harmonizes with the capabilities of current meta-analytical software, which is adept at delivering aggregated outcomes stemming from individual data. In light of your valuable feedback, we have carefully revised our manuscript. The text now clearly articulates that the Unit of Analysis shall be "all individuals included in the control and experimental groups", ensuring unambiguous communication of our research methodology (lines 385–386).

Comment 9. I am confused by your decision about the type of model you will be using – are you planning to split the studies between those that have ‘high’ heterogeneity vs ‘low’ heterogeneity? You will be looking for heterogeneity between studies, so how will you know which clusters of studies have high/low heterogeneity? I would suggest that you are unlikely to find studies with low heterogeneity, so a random-effects model for all of them would be more appropriate.

Response:

Thank you for your discerning comment regarding our proposed approach for addressing heterogeneity. As you insightfully pointed out, the anticipation of low heterogeneity among studies may be overly optimistic given the inherent variability in study designs and participant characteristics within the realm of group arts therapies.

To address this, we have made pertinent revisions to our manuscript. We explicitly describe how we will assess the statistical heterogeneity of included studies using the I^2 statistic and the Chi^2 test. Specifically, in our revised text on lines 393-395, we have delineated that studies will be considered exhibiting high heterogeneity if $I^2 > 50\%$ or $p < 0.1$. Furthermore, we state in lines 409-411 that for results from two or more studies with heterogeneity ($I^2 > 50\%$ or $p < 0.1$), we will combine them using a random effects model.

Comment 10. I would like to know more about why you have not involved patients or the public. It is best practice to seek input from the population you are writing about.

Response:

Thank you for raising the important issue of patient and public involvement in research.

In the context of our systematic review and meta-analysis, we focused on synthesizing published data from a range of sources to answer specific research questions regarding group arts therapies. This methodology traditionally does not include direct patient and public involvement, as it is based on a retrospective analysis of existing studies rather than on primary data collection or the design of new interventions.

However, we concur with your assertion that incorporating insights from those affected by the conditions under study can enrich the research process. Although direct involvement was not part of our systematic review protocol, we acknowledge that this component could play a significant role in the interpretation and dissemination phases. Therefore, while patients or the public were not directly involved in the conduct of our review, we plan to consult their opinions as we move forward with discussing the results in the future work.

Comment 11. In your discussion you write about MATISSE. You say it was a ‘systematic’ study, which is a bit confusing, as it was a trial not a review. You might want to adjust your language to make this clear. It was also only a trial of art therapy, not multiple arts therapies.

Response:

We appreciate your attention to detail and thank you for assisting us in enhancing the quality of our work. Your feedback has been invaluable in our revision process.

Upon revisiting the mentioned section, we see how our language may have inadvertently suggested that MATISSE was a systematic review rather than a clinical trial. In response to your feedback, we have revised lines 449-451 of our manuscript accordingly. The corrected text now reads: The landmark MATISSE study was a pivotal clinical trial focusing specifically on the use of art therapy for individuals with schizophrenia. It revealed that this singular approach within group arts therapies did not significantly enhance overall functioning or mental health outcomes, thus prompting further scrutiny into the effectiveness of group arts therapies for such conditions.

Comment 12. There is currently a large-scale trial of group arts therapies being undertaken in the UK

– the ERA study. There is a protocol paper which you might be interested in – Carr (2023) Effectiveness of group arts therapies (art therapy, dance movement therapy and music therapy) compared to group counselling for diagnostically heterogeneous psychiatric community patients: study protocol for a randomised controlled trial in mental health services (the ERA study).

Response:

Thank you for your insightful suggestions and for bringing to our attention this latest randomized controlled trial, which indeed represents a significant undertaking in the field of group art therapy. Following your recommendation, we have thoroughly reviewed the protocol paper by Carr (2023) and have incorporated a discussion of the study's design into our manuscript (lines 456 to 460). We share your enthusiasm for the potential contribution of the latest research to the field of group art therapy for people with schizophrenia. Accordingly, we acknowledge the importance of these impending outcomes and intend to discuss them in our work as soon as they are published.

Comment 13. There is a Cochrane review about art therapy for schizophrenia from 2005, one about music therapy from 2017, one about dance therapy from 2013, and one about dramatherapy from 2007. These should certainly be discussed.

Response:

Thank you for your valuable feedback. We recognize the significance of these seminal Cochrane reviews as cornerstones in the field that have indeed laid the groundwork for ongoing research. As you recommended, we have examined each of these foundational works and discussed their findings and implications in our manuscript from lines 461 to 486. In adding this content, our goal is to provide a comprehensive narrative that not only honors the established contributions of previous research but also emphasizes that our study aims to systematically evaluate the effectiveness of group-based artistic interventions within the clinical treatment for schizophrenia. Our findings intend to illuminate areas potentially underexplored in previous reviews and to contribute fresh insights into this area of therapeutic intervention.

VERSION 2 – REVIEW

REVIEWER	Khodaei-Ardakani, Mohammad-Reza University of Social Welfare and Rehabilitation Sciences, Psychiatry
REVIEW RETURNED	11-Mar-2024
GENERAL COMMENTS	In my opinion, according to the editing done, the article can be printed.